# Investigating the safety and activity of the use of BTT1023 (Timolumab), in the treatment of patients with primary sclerosing cholangitis (BUTEO): A single-arm, two-stage, open-label, multi-centre, phase II clinical trial protocol

Katherine Arndtz,[1,2] Margaret Corrigan,[1,2] Anna Rowe,[3] Amanda Kirkham,[4] Darren Barton,[3] Richard P Fox,[4] Laura Llewellyn,[5] Amrita Athwal,[3] Manpreet Wilkhu,[3] Yung-Yi Chen,[1,2] Chris Weston,[1,2] Amisha Desai,[6] David H Adams,[1,2] Gideon M Hirschfield,[1,2] on behalf of the BUTEO trial team

For numbered affiliations see end of article.

**Correspondence to**
Dr Gideon M Hirschfield; g. hirschfield@bham.ac.uk

## ABSTRACT

**Introduction** Primary sclerosing cholangitis (PSC) is a progressive inflammatory liver disease characterised by relentless liver fibrosis and a high unmet need for new therapies. Preventing fibrosis represents an important area of interest in the development of vital new drugs. Vascular adhesion protein-1 (VAP-1) drives inflammation in liver disease, and provision of an antibody against VAP-1 blunts fibrosis in murine models of liver injury.

**Methods and analysis** BUTEO is a single-arm, two-stage, open-label, multi-centre, phase II clinical trial. Up to 59 patients will receive treatment with anti-VAP monoclonal antibody, BTT1023, over a 78-day treatment period. Adults with PSC and a serum alkaline phosphatase (ALP) of at least 1.5 times the upper limit of normal will be included. Our primary outcome measure is a reduction in ALP by >25% from baseline to Day 99. Secondary outcome measures include safety and tolerability, changes pre therapy/post therapy in circulating serum VAP-1 as well as imaging findings. The first patient participant was recruited on 08 September 2015.

**Ethics and dissemination** This protocol has been approved by the Research Ethics Committee (REC, reference 14/EM/1272). The first REC approval date was 06 January 2015 with three subsequent approved amendments. This article refers to protocol V3.0, dated 16 March 2016. Results will be disseminated via peer-reviewed publication and presentation at international conferences.

**Trial registration** The trial is registered with the European Medicines agency (EudraCT: 2014-002393-37), the National Institute for Health Research (Portfolio ID: 18051) and ISRCTN: 11233255. The clinicaltrials.gov identifier is NCT02239211. Pre-results.

## Strengths and limitations of this study

► Unique, tailor-made clinical trial design incorporating a dose confirmatory and safety stage (based on the traditional 3+3 design), then followed by a phase II Simon's two-stage design.
► Brings the translation of laboratory research into a proof of activity clinical trial.
► An early-phase experimental medicine study of a novel first-in-class drug, in a chronic disease cohort with a large unmet need for new therapies.
► Aims to address not just the need for new therapies but also the need for reliable clinical trial endpoints as well as biomarkers for staging and predicting clinical outcomes.
► Small cohort due to primary sclerosing cholangitis being a rare orphan disease as well as unpredictability of the disease making stability for clinical trial inclusion difficult.
► Short duration of the treatment period in which goal is to demonstrate collective markers of efficacy to justify longer and placebo-controlled trials.
► Limited evidence base for the primary endpoint of a reduction in alkaline phosphatase in the context of anti-fibrotic agents, however accepting that there is no alternative surrogate currently available.
► Translational study from mice into human subjects and the unknown differences this may entail.

## INTRODUCTION

End-stage liver disease, regardless of aetiology, is characterised by progressive hepatic fibrosis culminating in liver cirrhosis and accompanying increased risks of liver cancer, liver failure, portal hypertension and death. Preventing progressive liver fibrosis represents an important area of interest in the development of new drugs suitable for all patients with liver disease. Primary sclerosing cholangitis (PSC) is a prime example

of a progressive inflammatory liver disease which is characterised by persistent liver fibrosis and a high unmet need for new therapies. PSC has a population incidence of 1.3 per 100 000 annually, with a prevalence of 16.2 per 100 000.[1 2 3] It affects both men and women, with a median age of 41 years,[4] and is associated with inflammatory bowel disease (IBD) in 80% of cases.[5] More than 50% of patients require liver transplantation within 10–15 years of symptomatic presentation,[6 7] reflecting the failure of medical therapies to have any impact on the clinical outcome: in the UK, for example, PSC is now the leading autoimmune liver disease indication for transplant, despite being the rarest of the autoimmune liver diseases. One barrier to the development of efficacious new medical therapies is the lack of clinically relevant endpoints and there is an urgent need to develop appropriate non-invasive surrogate endpoints to improve clinical trial design.[8]

### Vascular adhesion protein-1 (VAP-1)

Vascular adhesion protein-1 (VAP-1) is a 170-kDa homodimeric type 2 transmembrane sialoglycoprotein with a short cytoplasmic tail of no known signal sequence, a single transmembrane segment and a large extracellular domain. VAP-1 is constitutively expressed on human hepatic endothelium and supports lymphocyte adhesion and transendothelial migration. Cloning of VAP-1 revealed it to be a copper-dependent semicarbazide-sensitive amine oxidase (SSAO) which catalyses the oxidative deamination of exogenous and endogenous primary amines resulting in the generation of aldehyde, ammonia and $H_2O_2$. These products activate NF$\kappa$B-dependent chemokine secretion and adhesion molecule expression in liver endothelium and may initiate and propagate oxidative stress following the conversion of $H_2O_2$ to hydroxyl free radicals. A soluble form of VAP-1 (sVAP-1) accounts for nearly all of the circulating amine oxidase activity in humans.[9]

The progression of PSC to scarring, cirrhosis and hepatobiliary cancer is driven by a chronic inflammatory response and immune cell mediated destruction of bile ducts.[10] Our research implicates VAP-1 in the inflammation that drives fibrogenesis in liver disease.[11] VAP-1 also acts as an adhesion receptor to support leucocyte recruitment in liver inflammation, a function that is critical in animal models in the formation of fibrosis.[12] Thus, inhibition of VAP-1 is expected to impact both inflammation and fibrosis; indeed, treatment with an antibody against VAP-1 prevents fibrosis in murine models of liver injury.[9] Data also shows particularly high levels of circulating serum VAP-1 (sVAP-1) in patients with PSC as well as a strong correlation between sVAP-1/SSAO activity in serum and histological fibrosis scores in patients with fatty liver disease.[9] Based on the strong up-regulation of hepatic VAP-1 reported in PSC patients,[9] we hypothesise that levels of sVAP-1/SSAO will correlate with the severity of fibrosis in PSC and will predict patients at risk of progressive disease.

These observations underpin our proposal that VAP-1 has an important role in the progression of liver fibrosis. We now plan to test the hypothesis that inhibiting VAP-1 with a neutralising antibody (BTT1023) will reverse or delay fibrogenesis in patients with PSC. Additionally, reliable biomarkers that correlate with fibrosis stage and progression of liver disease are in demand in order to predict outcome and to stage disease, without the need for invasive liver biopsy. This research will allow us to translate laboratory research into a proof of activity clinical trial that will elucidate the role of VAP-1 in liver fibrosis and its potential as a therapeutic target and biomarker.

### BTT1023 (now known as Timolumab)

BTT1023 is a fully human, monoclonal, anti-VAP-1 antibody which blocks the adhesion function of VAP-1, thereby diminishing leucocyte entry into sites of tissue inflammation. Going forward, BTT1023 will be known as Timolumab; however, the original numeric name is used in this manuscript in order to remain in keeping with the trial protocol. In vivo, blocking VAP-1 function with an anti-mouse anti-VAP-1 antibody significantly alleviates inflammation in mouse models of arthritis and liver fibrosis.[13] BTT1023 appears to be safe and well tolerated in humans: BTT1023 has been given in doses up to 8 mg/kg in patients with rheumatoid arthritis and psoriasis after oral premedication (cetirizine and ibuprofen) and also appears safe and well tolerated in repeated intravenous dosing.[14] No cytokine release syndrome was seen. This dose therefore appears reasonable as a starting point and is considered to provide appropriate bridging into existing clinical data.

In the well-tolerated arthritis study, patients received 5 doses at fortnightly intervals with potentially efficacious trough levels being achieved by Day 42, after 3 doses. The psoriasis study used a more rapid loading of 3 doses at Days 1, 8 and 22, which was also well tolerated. Therefore, we have decided to deploy the accelerated dosing scheme (ie, dosing on Days 1 and 8 and biweekly thereafter) in the current study.[14]

BTT1023 is produced in a Chinese hamster ovary cell culture and purified with appropriate methods including specific viral inactivation and removal procedures. BTT1023 is contained in single-use 100 mg/10 mL glass vials and requires dilution with BTT1023 IV Infusion Diluent (0.9% sodium chloride and 0.02% polysorbate 80 in water for injection) prior to administration by intravenous infusion. The amount of diluent to be added to the BTT1023 concentrate is calculated to consistently provide a total diluted drug product infusion volume of 50 mL. BTT1023 and its diluent are stored at 2–8°C and the maximum shelf life for the diluted infusion solution is 24 hours when stored at this temperature.

### Objectives

The trial primary objective is to determine the activity of the anti-VAP-1 antibody BTT1023 in patients with PSC as measured by a decrease in alkaline phosphatase

(ALP) levels (primary endpoint) with secondary endpoints to include various measures of liver injury and fibrosis and evaluation of the safety, effective dose and tolerability of BTT1023 in patients with PSC. Our secondary objectives include determining the mechanisms of action of BTT1023 through in vitro assessment of sVAP-1 concentration, SSAO enzyme activity and immune cell function; evaluating the potential of a novel MRI-based assessment of liver fibrosis and biliary strictures for assessing therapeutic response in PSC; and assessing the use of sVAP-1/ SSAO as a biomarker to monitor disease progression in PSC.

## METHODS
### Study design overview
BUTEO is a two-stage single-arm, open-label, multi-centre hybrid trial of treatment with monoclonal anti-VAP-1 antibody, BTT1023, in adult patients with PSC. The sample size will be a maximum of 59 patients, who will each have up to 7 intravenous infusions of BTT1023 over a 78-day treatment period. All patients will be followed up until Day 120 (42 days after the last administration of treatment). At specified time points during each visit, serum will be taken for circulating levels of BTT1023 as well as for anti-drug antibodies, VAP-1 activity and additional exploratory research samples. This includes pre dose, during dose and post dose, with some patients also attending 24 hours later for further blood testing.

The trial is composed of two components. The run-in component of the trial incorporates a conventional 3+3 cohort design to confirm the therapeutic dose, with decisions regarding continuation based on toxicity and pharmacokinetic (PK) data, figure 1. The trial begins with the recruitment of 6 patients all receiving the starting dose of 8 mg/kg. Recruitment will be paused while awaiting the results of trough blood serum levels of circulating BTT1023 at Visit 7 (Day 50) from all 6 patients and until the dose-limiting toxicity (DLT) reporting period is completed for each patient (Visit 10 [Day 99]).

If results from the first cohorts show an acceptable DLT rate (see later) and trough levels of BTT1023 meet the stipulated success criteria, the trial will continue into the expansion component of the trial. Acceptable trough levels have been set a trough concentration of 3 µg/ mL free circulating BTT1023 at 8 weeks from first infusion, which is approximately 100-fold the dissociation constant ($K_d$) of BTT1023 from VAP-1 and will result in target occupancy of approximately 90%. In the event that DLT rate is acceptable but the PK levels do not meet the success criteria, then the trial moves into a conventional 3+3 cohort design, using escalating doses of BTT1023. In this event, the original cohort of 6 patients will no longer be evaluated, but a new cohort of 3 patients will be recruited to receive the newly identified test dose of 12 mg/kg. If there are no DLTs at Visit 10 [Day 99], an additional cohort of 3 patients will be recruited at the new test dose. If the DLT rate remains acceptable but the PK

levels still do not meet the success criteria, a further 3+3 patients will be recruited at the highest dose of 16 mg/kg. If this is found not to result in sufficient blood levels of BTT1023, then the trial will be stopped. If the PK values are found to be too high (such as resulting in trough levels consistently exceeding 100 µg/mL), then there is potential to de-escalate the dose in agreement with the Data Monitoring Committee (DMC) and after regulatory approvals. The trial will be stopped at any stage where patient safety is compromised. Individual patients will only receive one dose level. Once a confirmed dose has been established, the trial will expand until a total of 37 patients have received treatment with the confirmed dose, including those patients who have previously received this dose during the confirmatory period. Those patients not receiving the confirmed dose will not be included in the final evaluation.

### Patient selection
Patients with particularly elevated ALP levels are predicted to be more at risk of progressive disease; thus, these patients will be selected. Currently, four UK academic hospital centres are involved (based in Birmingham, Nottingham, Oxford and Newcastle) with further centres potentially coming on board shortly. Informed consent will be obtained by appropriately trained members of the research team at each site.

The main inclusion criteria are patients aged 18–75 years with a clinical diagnosis of PSC, as evidenced by chronic cholestasis of more than 6 months duration with either an MRI or liver biopsy consistent with PSC and in the absence of a documented alternative aetiology. Patients must have an ALP of at least 1.5 times the upper limit of normal. Those with concomitant IBD must have evidence to show clinically and colonoscopically stable disease within the past 12 months, without findings of high-grade dysplasia and without the need for biological therapies. In those on treatment with ursodeoxycholic acid (UDCA), therapy must be stable for at least 8 weeks prior to screening and at a dose not greater than 20 mg/ kg per day. In those not on treatment with UDCA at the time of screening, a minimum of 8 weeks since the last dose of UDCA should be recorded. As an IgG molecule, BTT1023 is not anticipated to interfere with chromosomal material; however, teratogenicity has not been studied and so rigorous steps will be taken to exclude pregnancy pre-treatment and with regard to need for effective contraception throughout the trial and for 99 days after. Some of the main exclusion criteria can be seen in box 1.

## TREATMENT
A single-arm, rather than placebo-controlled, design was chosen to allow efficient enrolment of patients into the study due to its intensive nature, in which a significant chance of being allocated into the placebo group may act as a substantial barrier for enrolment. The proposed

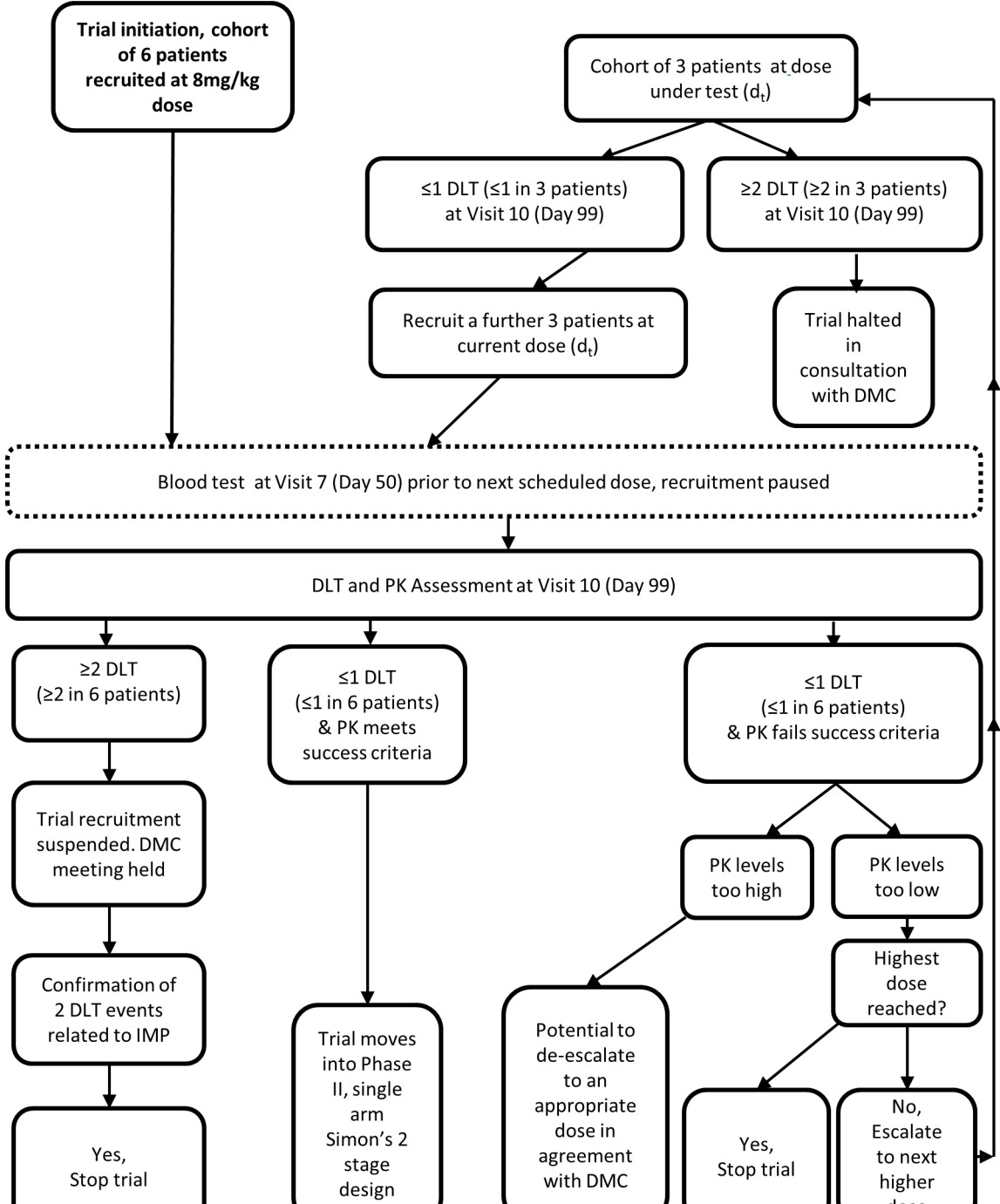

**Figure 1** Flow diagram showing the trial decision guidelines for the run-in period of the trial design, incorporating a conventional 3+3 cohort design if dose escalation is required.

primary endpoint (ALP) is a biochemical measurement and thus not open to subjective bias as a clinical assessment would be, rendering the necessity of a concurrent control group less vital.

## TRIAL SCHEMA

Given the unpredictable nature of PSC and natural variation of ALP levels, we have built into the trial a two-stage screening process over 4–7 weeks, whereby the ALP levels must not vary by more than 25% in order to

continue to enrolment. During screening, patients have routine blood screens and other non-invasive markers of liver fibrosis including Mayo PSC Risk Score, MELD, Enhanced Liver Fibrosis (ELF), Fibroscan and MRCP with additional LiverMultiscan imaging. These will then be repeated during treatment and in the follow-up period to assess for any change.

During all 7 treatment visits, patients will receive pre-medication with cetirizine 10 mg + ibuprofen 400 mg orally (in the absence of any contraindications)

**Box 1    Main exclusion criteria (not a complete list)**

- ► AST and ALT >10xULN or bilirubin >3xULN or INR >1.3 in the absence of anti-coagulants.
- ► Serum creatinine >130 µmol/L or platelet count <50 x 109/L.
- ► Any evidence of hepatic decompensation past or present, including ascites, hepatic encephalopathy or variceal bleeding.
- ► Recent cholangitis within last 90 days or ongoing need for prophylactic antibiotics.
- ► Pregnancy or breast feeding.
- ► Flare in colitis activity within last 90 days requiring intensification of therapy beyond baseline maintenance treatment; use of oral prednisolone >10 mg/day, biologics (ie, monoclonal antibodies) and or hospitalisation for colitis within 90 days. Prior use of biologics is not a contraindication to screening.
- ► Diagnosed cholangiocarcinoma or high clinical suspicion of cholangiocarcinoma.
- ► Concurrent malignancies or invasive cancers diagnosed within past 3 years except for adequately treated basal cell and squamous cell carcinoma of the skin and in situ carcinoma of the uterine cervix.
- ► Presence of a percutaneous drain, bile duct stent or prior organ transplantation.
- ► Participation in an investigational trial of a drug or device within 60 days of screening or 5 half-lives of the last dose of investigational drug.
- ► Positive screening test for tuberculosis (TB) (including T-SPOT.TB TB test), unless respiratory review confirms false-positive test results.
- ► Receipt of live vaccination within 6 weeks prior to baseline visit.

plus intravenous hydrocortisone 100 mg, 1–2 hours pre infusion (the latter for the first 3 doses only). The first infusion will be given over 2 hours, with a 4-hour monitoring period post infusion. Provided that no adverse reactions are seen, the infusion time will drop to 1 hour with initially a 3-hour observation period (for the second dose) and then down to 2 hours monitoring post infusion (for all subsequent doses). Safety investigations will be completed pre infusion and post infusion and they include haematology/biochemistry sampling, along with ECG, clinical assessment and physical examination. The full trial schema can be seen as figure 2.

A aliquot (0.5–1.0 mL) of the BTT1023 infusion solution will be taken at the end of every infusion and refrigerated. These samples may be used for analysis of BTT1023 concentration if anomalies in PK data are observed that could be due to errors in IMP preparation.

### Data analysis plan
#### Power calculations and sample size justification
The dose confirmatory stage requires a maximum of 18 patients. The sample size has been based on the classic 3+3 design, investigating three fixed dose increments (8, 12 and 16 mg/kg), with no dose skipping.

The expansion phase follows a Simon's two-stage minimax design with lower and upper acceptability bounds of 15% and 30%, respectively, and error rates $\alpha=0.10$ and $\beta=0.20$. Thirty-seven patients are required in this stage of the trial; however, to account for patient dropout, estimated at approximately 10%, the sample size is extended by a further 4 patients. As such, the target recruitment is up to 41 patients for the expansion phase and up to 59 participants in total. An interim assessment will examine the primary outcome once 18 evaluable patients have been recruited at the confirmed dose. If 3 or more responses are observed in stage 1, then the trial will continue into stage 2. Recruitment will not be halted while stage 1 is assessed. If the stage 1 criterion is not met, then the trial will cease. However, if the criterion is met, then further patient recruitment continues until 37 evaluable patients are recruited. If overall there are 9 or more responses from 37 evaluable patients, then we conclude that the treatment warrants further investigation.

In this setting, the interpretation of alpha ($\alpha$) is the probability satisfying stage 1 and observing 9 or more responses in 37 patients overall when the true response rate is 15%; a false-positive result (type 1 error). Beta ($\beta$) is the probability of failing to acknowledge activity when the true response rate is 30% (type 2 error). As such, the power, $1-\beta$, is the probability of taking an effective treatment forward.

### OUTCOME MEASURES AND ANALYSIS
This is an early-phase trial of BTT1023 in immune cell mediated liver disease, with the rationale to identify biochemical efficacy (reduction in ALP) and safety, in an orphan disease that presently lacks any other effective medical therapy. The trial design therefore focuses on identifying early biochemical efficacy signals to justify larger-scale, randomised controlled trials of a longer duration.

Our primary outcome measure is patient response to treatment at Day 99, and it is a reduction in serum ALP by 25% or more from baseline to Day 99. Our data on stability of ALP in PSC suggests that such responses occur very seldom during the natural course of the disease, and we can therefore reliably assess changes from baseline and response rates for this proof of concept study to evaluate the therapeutic potential of BTT1023. In addition, we are excluding patients in whom the levels change significantly naturally by >25% in the 5-week to 7-week screening period.

Secondary outcome measures include safety and tolerability via treatment compliance, patient withdrawal, frequency of serious adverse event (SAE)/adverse event (AE), change in quality of life questionnaires (EQ-5D, Fatigue Severity Scale, Pruritus Visual Analogue Score, Inflammatory Bowel Disease Diaries) and change in quality of BTT1023 efficacy (via tests of liver fibrosis including ELF, Fibroscan and liver biochemistry). Additionally, liver MRI is an emerging method for monitoring liver disease and its treatment. We will evaluate changes in MRI imaging pre therapy and post therapy using the LiverMultiscan protocol (or equivalent methodology, at sites where this is possible). Finally, we will evaluate changes in sVAP-1/SSAO as a biomarker of liver disease activity across the study period.

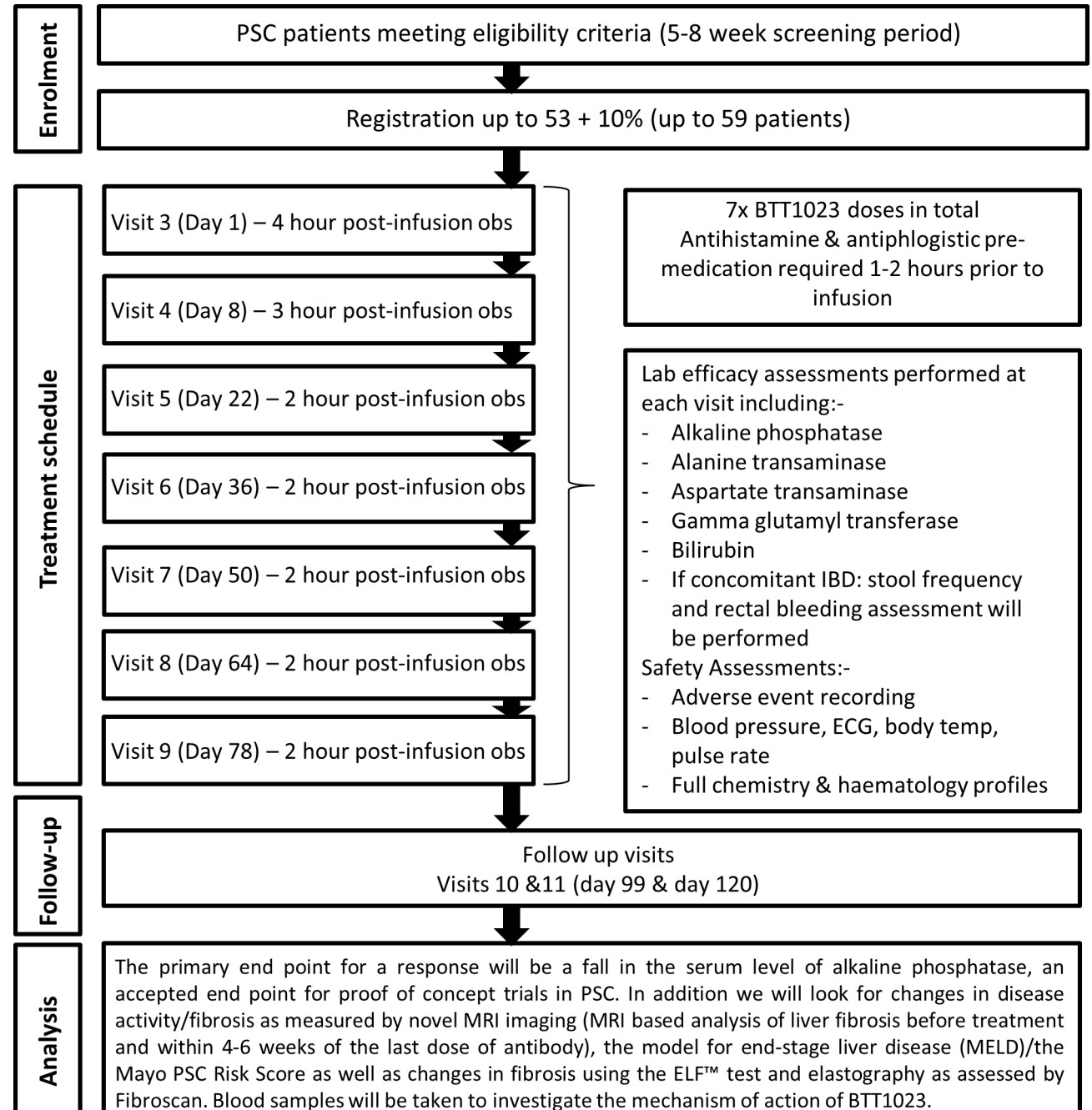

**Figure 2**   BUTEO trial schema.

## INTERIM ANALYSIS

An interim analysis will be completed once 18 patients have been evaluated for the primary outcome (ALP response). If 3 or more responses are observed, then the trial will continue. If this is not met, then the trial will cease. If adequate response is seen, a further 19 patients will be recruited in order to obtain the required sample size of 37 patients (allowing for 10% patient dropout during trial duration, this number could reach a total of 41 patients recruited). The final success criterion chosen maintains power and, in doing so, can incur an increased type 1 error rates (note that the stage 1 criterion is fixed since the final sample size is unknown at stage 1). These

have been calculated by the trial biostatistician and independently verified by a Cancer Research UK Clinical Trials Unit biostatistician.

Patients that cannot be evaluated for the primary outcome (eg, due to withdrawal or lost to follow-up) will be treated as non-responders.

### Final analyses

Final analyses of the primary outcome will be performed when all patients have been followed to Day 120 and once the database has been locked. If overall there are 9 or more responses from 37 evaluable patients, then we conclude that the treatment warrants further investigation. Only

patients treated at the confirmed dose will contribute to the total patient requirement.

## CONDUCT OF THE TRIAL

This is a clinician-initiated and clinician-led trial funded by the National Institute for Health Research (NIHR) which receives funds directly from the Department of Health. Biotie Therapies will be supplying BTT1023 free of charge to all individual NHS trusts that will be directly treating patients as part of this clinical trial. There are payments to individual NHS trusts on a per-patient basis, to cover trial running costs. The University of Birmingham is acting as trial sponsor and, as such, remains responsible for the study conduct.

## ADVERSE EVENT REPORTING
### Dose-limiting toxicity

Dose-limiting toxicity (DLT) is defined as an AE that meets the criteria of grade 3 cytokine release syndrome or grade 4/5 for any criteria, as defined in the Common Terminology Criteria for Adverse Events (CTCAE V4.0). Although previous studies have shown no DLTs with BTT1023, toxicity monitoring will be ongoing throughout the trial and any concerns will be reported to the Trials Office within 24 hours of the investigator becoming aware of the event. The DLT reporting period is from Visit 3 (first infusion) to Visit 10 (21 days after last infusion). An acceptable DLT rate has been established for the trial as a maximum of 1 incidence in 6 patients (~17%). If the DLT rate rises to 2 or more, at any stage during the DLT reporting period (Visit 3 [Day 1] to Visit 10 [Day 99]), the trial will be halted in consultation with the DMC.

### Adverse events/serious adverse events

The collection and reporting of AEs will be in accordance with the Medicines for Human Use Clinical Trials Regulations 2004 and its subsequent amendments. The CTCAE V4.0 criteria will be used to grade each AE. Any pre-existing conditions will be reported in the medical history and will not be reported as an AE unless the condition worsens by at least one CTC grade during the trial. The reporting period for AEs will commence from the date of consent (Visit 1) and will continue until the final follow-up visit (Visit 11: Day 120) or alternatively up to 45 days post last infusion if the patient withdraws from the study prior to completion of all 7 study drug infusions. All trial patients will continue to receive standard concomitant clinical care throughout the study.

The Sponsor, appropriate regulatory authority (eg, Medicines and Healthcare Products Regulatory Agency (MHRA)) and the Research Ethics Committee (REC) will be informed of all SAEs as required by current regulations. An SAE judging to have a reasonable causal relationship to the drug will be recorded as a serious adverse reaction or a suspected unexpected serious adverse reaction, as appropriate, and will be reported to the MHRA and REC

within 7 days. The independent DMC will also review all SAEs.

In the event that a patient or their partner becomes pregnant during the SAE reporting period, this will be recorded and reported and followed up, subject to required patient/partner approvals.

## MONITORING

Independent onsite monitoring will be carried out as required following an initial site-specific risk assessment. Additional on-site monitoring visits will be triggered, for example, by poor Case Report Form (CRF) return, poor data quality, low SAE reporting rates, excessive number of patient withdrawals or deviations. Any major problems identified during monitoring may be reported to the Trial Management Group and the Trial Steering Committee and the relevant regulatory bodies.

## DISCUSSION

PSC is a prime example of a progressive inflammatory liver disease characterised by relentless liver fibrosis. There is a high unmet need for new therapies as no currently licensed therapy has been shown to alter the natural course of the disease. The progression of PSC to scarring, cirrhosis and hepatobiliary cancer is driven by a chronic inflammatory response and immune cell mediated destruction of bile ducts. Our research suggests that VAP-1 is heavily implicated as a key driver for fibrogenesis and, as such, it provides a target for the possible slowing or even reversal of the liver damage seen in PSC.

The unpredictability of PSC along with its designation as a rare orphan disease pose particular challenges in trial design. ALP fluctuates during the natural course of the disease which limits its usefulness as a primary endpoint; however, it is commonly used as a standard marker of PSC disease activity, in the absence of a viable alternative. Thus, reliable biomarkers that correlate with fibrosis stage and progression of liver disease are in demand in order to predict outcome and to stage disease without the need for invasive liver biopsy, and this trial will aid in investigating the role of VAP-1 in liver fibrosis and its potential as a therapeutic target and biomarker.

This unique trial design, incorporating a dose confirmatory and safety stage (based on the traditional 3+3 design), then followed by a phase II Simon's two-stage design is aimed to determine a safe and well-tolerated dose of BTT1023 and the efficacy of this treatment in a new disease group.

Screening for this study commenced on 01 February 2015 with the first patient registered on 10 September 2015. Recruitment is ongoing.

### Ethics and dissemination

The trial is being performed in accordance with the recommendations guiding physicians in biomedical research involving human subjects, adopted by the 48th World Medical Association General Assembly as well as

the Research Governance Framework for Health and Social Care, the applicable UK Statutory Instruments (which include the Medicines for Human Use Clinical Trials 2004 and subsequent amendments and the Data Protection Act 1998) and Guidelines for Good Clinical Practice. The protocol has been approved by the REC with the reference of 14/EM/1272. The first REC approval date was 06 January 2015 with subsequent amendments on 18 March 2015 (non-substantial), 27 November 2015 (substantial) and 16 March 2016 (substantial) also now approved. All active sites have obtained local Research and Development department approval and are up to date with the latest protocol amendment.

Standard regulations for data handling and patient confidentiality are being followed, in accordance with the 1998 UK Data Protection Act. During the trial, patients will be identified using only their unique trial number, initials and date of birth on the CRF and in correspondence between the Trials Office and the participating site. Data quality will be maintained according to recognised guidance and will be consistent to the source data. All essential trial documentation and source records will be securely retained for at least 15 years after the end of the trial or following the processing of all biological material collected for research, whichever is the latter. Results will be disseminated via peer-reviewed publication and presentation at international conferences; additional summaries will be provided to patients and patient support groups.

**Author affiliations**
[1]Centre for Rare Diseases, Institute of Translational Medicine, Birmingham Health Partners, University Hospitals Birmingham, Birmingham, UK
[2]NIHR Birmingham Biomedical Research Centre, Centre for Liver Research, University of Birmingham, Birmingham, UK
[33]NIHR Birmingham Biomedical Research Centre, Clinical Trials Group (D3B Team), CRUK Clinical Trials Unit, University of Birmingham, Birmingham, UK
[4]Department of Statistics, CRUK Clinical Trials Unit, University of Birmingham, Birmingham, UK
[5]Early Drug Development Team, CRUK Clinical Trials Unit, University of Birmingham, Birmingham, UK
[6]Pharmacy, University Hospitals Birmingham NHS Foundation Trust, Queen Elizabeth Hospital Birmingham, Birmingham, UK

**Acknowledgements** This project is funded by the Efficacy and Mechanism Evaluation (EME) Programme*, an MRC and NIHR partnership. The authors would like to thank the NIHR, the NIHR BRC in Birmingham, Biotie Therapies, the Queen Elizabeth Hospital Liver PPI group and the patients involved. *The EME Programme is funded by the MRC and NIHR, with contributions from the CSO in Scotland and NISCHR in Wales and the HSC R&D Division, Public Health Agency in Northern Ireland. This research is supported by the National Institute For Health Research (NIHR) Birmingham Liver Biomedical Research Centre (BRC). This paper presents independent research and the views expressed are those of the author(s) and not necessarily those of the NHS, the NIHR or the Department of Health.

**Contributors** All authors have read and approved the final manuscript. KA is lead sub-investigator at the Birmingham BUTEO site and wrote this manuscript in conjunction with GH. CW, MC, AR, MW, RPF, AK, DB, DHA and GH were involved in the protocol development. RPF and AK also contribute statistical analysis. DHA was responsible for the original concept.

**Funding** This work is supported by the NIHR Efficacy and Mechanism Evaluation (EME) Programme, grant number 12/165/31. The full title is: "Targeting vascular adhesion protein-1 (VAP-1) for the treatment of liver fibrosis: a study of efficacy and mechanisms in patients with primary sclerosing cholangitis (PSC)". This article represents independent research funded by the National Institute for Health Research (NIHR) Birmingham Liver Biomedical Research Unit (BRU).

**Disclaimer** The views expressed in this publication are those of the author(s) and not necessarily those of the MRC, NHS, NIHR or the Department of Health.

**Competing interests** CW and DHA report grants from NIHR MRC, during the conduct of the study; grants and non-financial support from Pharmaxis Inc, Australia, outside the submitted work; and have a patent "The use of VAP-1 inhibitors for treating fibrotic conditions (WO 2011029996 A1)" issued. Other authors have no competing interests declared.

**Ethics approval** Research Ethics Committee.

**Provenance and peer review** Not commissioned; externally peer reviewed.

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
