## [Reviewer comments · BMJ Open]

ARTICLE DETAILS

TITLE (PROVISIONAL)	Investigating the safety and activity of the use of BTT1023 (Timolumab), in the treatment of patients with primary sclerosing cholangitis (BUTEO): A single arm, two-stage, open-label, multi-centre, phase II clinical trial protocol.
AUTHORS	Arndtz, Katherine; Corrigan, Margaret; Rowe, Anna; Kirkham, Amanda; Barton, Darren; Fox, Richard; Llewellyn, Laura; Athwal, Amrita; Wilkhu, Manpreet; Chen, Yung-Yi; Weston, Chris; Desai, Amisha; Adams, David; Hirschfield, Gideon

VERSION 1 - REVIEW

REVIEWER	Pietro Invernizzi Program for Autoimmune Liver Diseases, Section of Digestive Diseases, International Center for Digestive Health, Department of Medicine and Surgery, University of Milan-Bicocca.
REVIEW RETURNED	12-Dec-2016

GENERAL COMMENTS	This paper by Arndtz et al report a first attempt to develop Timolumab as a novel drug for primary sclerosing cholangitis. There is a great need for novel therapies for this disease. The group is well known and active in the field and this attempt is of great value. The study is well designed, performed and written. Further studies are still needed.
---

REVIEWER	Cyriel Y. Ponsioen Academic Medical Center, Amsterdam The Netherlands Dr. Ponsioen has received grant support from Takeda, and consultancy and speaker's fees from Takeda, Abbvie, and Dr. Falk Pharma
REVIEW RETURNED	23-Jan-2017

GENERAL COMMENTS	Minor comments: On page 7 the VAP1/SSAO ratio is introduced without any explanation. Page 8: How many patients have been exposed to timolimimumab in the arthritis and psoriasis studies? an arrow seem to be missing in figure 1 page 17-20 are very important for IRB purposes but can be largely condensed for the readers of BMJ Open
---

REVIEWER	Dr SM Rushbrook Norfolk and Norwich University Hospitals NHS Foundation Trust. United Kingdom
REVIEW RETURNED	19-Feb-2017

GENERAL COMMENTS	The above study will seek to develop a new therapy in Primary Sclerosing Cholangitis. As the authors point out this is an area where there is a large unmet clinical need and their choice of a biological target is scientifically sound, based on a large body of work from this group. The trial design is being conducted in a logical way and has a clear end point. As with all studies in PSC, the authors have had the difficulty in deciding what endpoint to use. They have chosen Alkaline phosphatase which has been endorsed by the PSC community, but clearly has its own limitations. The authors acknowledge this and try to allow for this. It would be useful in the protocol to say that a patient could not enter if they have had a biliary endoscopic procedure with in 3 months of enrollment in addition to the other inclusion/exclusion criteria set out. Clearly there is a risk in PSC that their therapy good be having a antifibrotic effect and yet not have an impact on cholestasis per say. After all, these 2 things do not always go hand in hand. I would therefore suggest that the authors try to define secondary end points of efficacy with regards to fibrosis, that if detected, could alter how a further study was done with regards to end point design in the future if the study doesn't meet its primary endpoint.
--

VERSION 1 – AUTHOR RESPONSE

Reviewer 1

Many thanks. There are no changes required.

Reviewer 2

1) "On page 7 the VAP1/SSAO ratio is introduced without any explanation".

Thank you; our omission, in an effort to keep the manuscript brief we failed to provide sufficient background. We now add at the start of the VAP-1 section:

Vascular adhesion protein-1 (VAP-1) is a 170-kDa homodimeric type 2 transmembrane sialoglycoprotein with a short cytoplasmic tail of no known signal sequence, a single transmembrane segment, and a large extracellular domain. VAP-1 is constitutively expressed on human hepatic endothelium and supports lymphocyte adhesion and transendothelial migration. Cloning of VAP-1 revealed it to be a copper-dependent semicarbazide-sensitive amine oxidase (SSAO) which catalyzes the oxidative deamination of exogenous and endogenous primary amines resulting in the generation of aldehyde, ammonia, and H₂O₂. These products activate NFκB-dependent chemokine secretion and adhesion molecule expression in liver endothelium and may initiate and propagate oxidative stress following the conversion of H₂O₂ to hydroxyl free radicals. A soluble form of VAP-1 (sVAP-1) accounts for nearly all of the circulating amine oxidase activity in humans.

2) "Page 8: How many patients have been exposed to timolimumab in the arthritis and psoriasis

studies?”

Thank you. This information is available in the IMPD which has been provided to the MHRA and the ethics board. As per the IMPD Biotie has prior to this study conducted three phase 1, placebo-controlled clinical studies with BTT1023 in which a total of 72 individual subjects have received antibody at various doses. The highest dose studied to date has been 8 mg/kg administered bi-weekly on five repeated occasions by i.v. infusion. The three studies were:-

i) First-in-man in healthy male volunteers conducted in the UK (BTT12-CD012; EudraCT Number: 2006-006610-16); total N on active=29.

ii) Phase 1b multiple ascending dose study in patients with plaque psoriasis conducted in Germany (BTT12-CD016; EudraCT Number: 2008-004209-32); total N on active=23.

iii) Phase 1b multiple ascending dose study in patients with rheumatoid arthritis conducted in Bulgaria (BTT12-CD015; EudraCT Number 2008-005721-10); total N on active=20.

3) “An arrow seems to be missing in figure 1”.

Many thanks – this has been replaced. Changes are on page 28 and below:

4) “Page 17-20 are very important for IRB purposes but can be largely condensed for the readers of BMJ Open”.

We have abridged these sections as requested. Changes are on pages 17-19 and the abbreviated text shown below:-

CONDUCT OF THE TRIAL

This is a clinician-initiated and clinician-led trial funded by the National Institute for Health Research (NIHR) which receives funds directly from the Department of Health (DoH). Biotie Therapies will be supplying BTT1023 free of charge to all individual NHS trusts that will be directly treating patients as part of this clinical trial. There are payments to individual NHS trusts on a per-patient basis, to cover trial running costs. The University of Birmingham are acting as trial sponsor and as such remains responsible for the study conduct.

ADVERSE EVENT REPORTING

Dose Limiting Toxicity (DLT) is defined as an Adverse Event (AE) that meets the criteria of grade 3 cytokine release syndrome or grade 4/5 for any criteria, as defined in the Common Terminology Criteria for Adverse Events (CTCAE V4.0). Although previous studies have shown no DLTs with BTT1023, toxicity monitoring will be ongoing throughout the trial and any concerns will be reported to the Trials office within 24 hours of the investigator becoming aware of the event. The DLT reporting period is from Visit 3 (first infusion) to Visit 10 (21 days after last infusion). An acceptable DLT rate has been established for the trial as a maximum of 1 incidence in 6 patients (~17%). If the DLT rate rises to 2 or more, at any stage during the DLT reporting period, (Visit 3 [Day 1] to Visit 10 [Day 99]) the trial will be halted in consultation with the DMC.

AEs/SAEs

The collection and reporting of Adverse Events (AEs) will be in accordance with the Medicines for Human Use Clinical Trials Regulations 2004 and its subsequent amendments. The CTCAE version

4.0 criteria will be used to grade each AE. Any pre-existing conditions will be reported in the medical history, and will not be reported as an AE unless the condition worsens by at least one CTC grade during the trial. The reporting period for AEs will commence from the date of consent (Visit 1) and will continue until the final follow-up visit (Visit 11: Day 120), or alternatively up to 45 days post last infusion if the patient withdraws from the study prior to completion of all 7 study drug infusions. All trial patients will continue to receive standard concomitant clinical care throughout the study.

The Sponsor, appropriate regulatory authority (e.g. Medicines and Healthcare Products Regulatory Agency (MHRA)) and the Research Ethics Committee (REC) will be informed of all SAEs as required by current regulations. An SAE judging to have a reasonable causal relationship to the drug will be recorded as a Serious Adverse Reaction (SAR) or Suspected Unexpected Serious Adverse Reaction (SUSAR), as appropriate and will be reported to the MHRA and REC within 7 days. The independent Data Monitoring Committee (DMC) will also review all SAEs.

In the event that a patient or their partner becomes pregnant during the SAE reporting period this will be recorded and reported and followed up, subject to required patient/partner approvals.

MONITORING

Independent onsite monitoring will be carried out as required following an initial site-specific risk assessment. Additional on-site monitoring visits will be triggered for example by poor CRF return, poor data quality, low SAE reporting rates, excessive number of patient withdrawals or deviations. Any major problems identified during monitoring may be reported to the Trial Management Group and the Trial Steering Committee and the relevant regulatory bodies.

Reviewer 3

1) "It would be useful in the protocol to say that a patient could not enter if they have had a biliary endoscopic procedure within 3 months of enrolment in addition to the other inclusion/exclusion criteria set out".

Many thanks; apologies if we were not clear enough but in the current protocol, having undergone a biliary endoscopic procedure within 3 months of enrolment is not an automatic exclusion to entry into this study, as long as no drain or biliary stent is currently in situ, there has been no recent cholangitis and there is no clinical suspicion of cholangiocarcinoma. These three circumstances are exclusion criteria and are thus already included within Table 1.

2) "Clearly there is a risk in PSC that their therapy could be having an antifibrotic effect and yet not have an impact on cholestasis per say. After all, these 2 things do not always go hand in hand. I would therefore suggest that the authors try to define secondary end points of efficacy with regards to fibrosis, that if detected, could alter how a further study was done with regards to end point design in the future if the study doesn't meet its primary endpoint".

Thank you. We of course agree. We have elected not to define all the research end-points in this manuscript, but as the reviewer very aptly notes better markers of fibrosis turnover are key. To that end we are exploring whether turnover assays, as available using the Nordic Bioscience fibrosis markers, can in particular contribute to the experimental research aspects of our study, and hope to report this data in the fullness of time.